# Metabolomics-Based Studies Assessing Exercise-Induced Alterations of the Human Metabolome: A Systematic Review

**DOI:** 10.3390/metabo9080164

**Published:** 2019-08-09

**Authors:** Camila A. Sakaguchi, David C. Nieman, Etore F. Signini, Raphael M. Abreu, Aparecida M. Catai

**Affiliations:** 1Physical Therapy Department, Federal University of São Carlos, São Carlos, SP 13565-905, Brazil; 2North Carolina Research Campus, Appalachian State University, Kannapolis, NC 28081, USA

**Keywords:** exercise, sports, metabolomics, metabolism

## Abstract

This systematic review provides a qualitative appraisal of 24 high-quality metabolomics-based studies published over the past decade exploring exercise-induced alterations of the human metabolome. Of these papers, 63% focused on acute metabolite changes following intense and prolonged exercise. The best studies utilized liquid chromatography mass spectrometry (LC-MS/MS) analytical platforms with large chemical standard libraries and strong, multivariate bioinformatics support. These studies reported large-fold changes in diverse lipid-related metabolites, with more than 100 increasing two-fold or greater within a few hours post-exercise. Metabolite shifts, even after strenuous exercise, typically return to near pre-exercise levels after one day of recovery. Few studies investigated metabolite changes following acute exercise bouts of shorter durations (< 60 min) and workload volumes. Plasma metabolite shifts in these types of studies are modest in comparison. More cross-sectional and exercise training studies are needed to improve scientific understanding of the human system’s response to varying, chronic exercise workloads. The findings derived from this review provide direction for future investigations focused on the body’s metabolome response to exercise.

## 1. Introduction

Acute and chronic physical activity causes extensive adaptations in organs and systems, leading to health benefits [1]. Improvements in technology have allowed investigators to quantify these adaptations using a biological systems approach, overlaying gene information with transcriptomics, proteomics, and metabolomics [1,2,3,4,5,6,7,8,9,10]. Combined data from multi-omics approaches will improve scientific understanding regarding the complex modulating effect that physical activity has on the phenotype at the individual level and related molecular mechanisms.

Metabolomics is defined as the simultaneous measurement of numerous low molecular metabolites that participate as substrates, reactants, signaling agents, intermediates, and products of enzyme-mediated reactions [3,4]. Metabolites are the final endpoints of upstream biochemical processes, and closely reflect the expressed phenotype. With the support of advanced analytical platforms and bioinformatics, metabolomics data can provide valuable insights regarding the biological impact of physical activity, pharmacological treatment, nutritional interventions, and other exposures [3]. 

Global metabolomics procedures were first performed in the 1960s and 1970s when gas chromatography mass spectrometry (GC-MS) was used to measure human metabolites in blood and urine samples [4]. Despite this, metabolomics was considered an emerging field of scientific endeavor as late as 2010, the year when the earliest studies investigating exercise effects in human athletes were published [3]. Since then, a growing number of research groups have used metabolomics in exercise-based studies. This is due, in large part, to the widespread availability of mass spectrometry platforms, freely accessible online databases of metabolites such as the Human Metabolome Database (HMDB), the expansion of chemical standards libraries, and advanced bioinformatics support to analyze and make sense of the large volumes of data. The net effect has been an improved capacity to accurately detect a greater number of metabolites and then interpret the overall effect on the human metabolome in a wide variety of matrixes. 

This systematic review provides a qualitative appraisal of metabolomics-based studies published during the past decade exploring exercise-induced alterations on the human metabolome. The conclusions derived from this review will provide an evidence-based framework for future investigations.

## 2. Results

A total of 1355 articles were retrieved for this analysis. Of these, 1314 were excluded for not meeting analysis criteria after review of the abstracts. Of the 41 studies selected for full text examination, six were excluded for not meeting analysis criteria. Of the 35 studies included for scoring, 24 achieved a minimum score of 6, and were included in the final analysis (Table 1 and Figure 1).

### 2.1. Exercise Intensity and Duration Effects on Metabolism 

Metabolic responses to exercise depend on the intensity and duration of effort. For the purposes of this review, heavy and moderate-intensity were differentiated using an intensity threshold of 60% of the oxygen uptake and heart rate reserve, and long and short-duration using a duration threshold of 60 min [40]. 

### 2.2. High-Intensity and Long-Duration 

More than half of the studies included in this analysis (62.5%; *n* = 15) measured metabolite responses to long-duration, high-intensity running (*n* = 8) [8,9,12,13,14,19,22,24], cycling (*n* = 5) [5,7,17,18,20], soccer (*n* = 1), and swimming (*n* = 1) [16] (Table 2). Liquid chromatography mass spectrometry (LC-MS) with or without GC-MS was used for metabolite identification in 11 [5,7,8,9,12,13,14,17,19,20,24] of these studies, with GC-MS as the primary method in two studies [16,22], capillary electrophoresis time-of-flight mass spectrometry (CE-TOFMS) in one study [21] and nuclear magnetic resonance (NMR) for one study [18]. Large-fold changes in metabolites from the lipid super pathway were reported by most investigators, including increases in plasma medium- and long-chain fatty acids, fatty acid oxidation products (dicarboxylate and monohydroxy fatty acids, acylcarnitines), and ketone bodies, with corresponding decreases in triacylglycerol esters (Table 2). Other metabolite changes included shifts in amino acids and increases in energy tricarboxylic acid (TCA) cycle components. 

### 2.3. High-Intensity and Short-Duration, Moderate-Intensity and Short/Long-Duration, Cross-Sectional and Training Studies

(A) High-Intensity, Short-Duration

Two studies measured metabolite responses to high-intensity, short-duration (18 to 30 min) exercise in recreationally active males and soccer athletes [23,28] (Table 3). Metabolite data from these studies were derived from GC-MS and NMR analytical platforms. Relatively small post-exercise changes were reported for metabolites related to the TCA cycle and related bioenergetics pathways. 

(B) Moderate-Intensity, Short-Duration 

One study reported metabolite responses following moderate-intensity, short-duration (30 min) cycling [6] (Table 3). Metabolites were identified using GC-MS and LC-MS/MS analytical platforms. Small-fold post-exercise changes were reported for metabolites linked to energy metabolism (lipolysis, glycolysis, TCA cycle intermediates, and catecholamines). 

(C) Moderate-Intensity, Long-Duration 

Two studies investigated metabolite responses to moderate-intensity, long-duration cycling and cross-country skiing [10,11]. Small to moderate post-exercise changes were reported for metabolites related to glycolytic and lipid pathways including free fatty acids, branched chain amino acids, acylcarnitines, mono- and diacylglycerols, and TCA intermediates.

(D) High- and Moderate-Intensity, Long-Duration

One study compared metabolite responses of high-intensity interval training (HIIT) and 60 min of moderate-intensity cycling (MOD) using GC-MS [26]. Small- to moderate-fold changes were reported for metabolites related to energy metabolism, and glycolytic and lipid pathways. HIIT compared to MOD induced higher post-exercise levels for glycolytic-related metabolites, with lower levels of lipid related metabolites. 

(E) Cross-Section Elite Athletes 

One study compared plasma metabolite levels in athletes from high- (*n* = 121) and moderate- (*n* = 70) endurance sports [15] (Table 3). Metabolomics was performed by ultra-performance liquid chromatography mass spectrometry (UPLC-MS/MS). The cross-sectional analysis showed some group differences, including higher levels for metabolites related to oxidative stress, fatty acid metabolism, steroid biosynthesis, and energy metabolism in high power and high endurance athletes. Plasma levels of metabolites related to steroid and polyamine pathways were more prominent in endurance athletes, with sterols, adenine-containing purines, and energy metabolites more evident in power athletes. 

(F) Chronic Training, Low-, Moderate-, and High-Intensity 

One study compared the chronic effects of cycling training at different intensities [25]. Plasma metabolites were measured with NMR, and only small group differences were reported in a few selected metabolites (hippuric acid, hypoxanthine, creatinine, dimethylamine, 3-methylxanthine).

(G) Chronic Training, High-Intensity

One study investigated the effects of chronic, high-intensity, short-duration running on plasma metabolite levels using NMR [27]. Small changes in selected metabolites were reported including lactate, pyruvate, TCA intermediates, and phospholipids.

## 3. Discussion

Advances in mass spectrometry since 2010 have led to an increasing number of metabolomics-based studies targeted on whole-body metabolite responses to varying acute and chronic exercise workloads. This systematic review of 24 high-quality papers published during the past decade revealed that the primary focus (63% of studies) has been on acute metabolite perturbations to long-duration, high-intensity aerobic exercise. Little information is available regarding metabolite changes coupled to acute bouts of exercise with lower workload volumes or those linked to long-term exercise training. The best studies utilized LC-MS/MS analytical platforms with large chemical standards libraries to identify and detect exercise-induced shifts in hundreds of metabolites. Strong bioinformatics support has improved predictive and descriptive modelling, discriminative variable selection, and the overall understanding of the body’s metabolome response to exercise.

This review indicates that a bout of prolonged and intensive exercise causes large-fold changes in numerous and diverse lipid-related metabolites [5,7,8,9,12,14,17,19,20,21,22,24]. In a typical study with human athletes exercising intensely for more than two hours, significant increases in at least 300 identified metabolites can be measured by LC-MS/MS analytical platforms, with more than 100 increasing twofold or greater [5,7,14,20,24]. This response includes post-exercise increases in plasma medium- and long-chain fatty acids, ketone bodies, fatty acid oxidation products, and sulfated bile acids. At the same time, related decreases occur in plasma triacylglycerol esters, primary and secondary bile acids, and minor phospholipids such as lysophosphatidylcholines and lysophosphatidylethanolamines [12,19,20,41]. Untargeted metabolomics has revealed post-exercise increases in both common (e.g., oleate/vaccinate, palmitate, linoleate, stearate, palmitoleate, myristate), and atypical fatty acids (adrenate, docosapentaenoate, dihomo-linolenate, dihomolinoleate, docosadienoate, and eicosenoate). The corresponding fatty acid oxidation signature includes acylcarnitines, 3-hydroxybutyrate (BHBA), and dicarboxylate and monohydroxy fatty acids. Other important shifts have been measured for plasma concentrations of tryptophan- and other amino acid-related metabolites, and energy tricarboxylic acid (TCA) cycle components including malate, aconitate, citrate, fumarate, succinate, and alpha-ketoglutarate [13,16,17,18,19,22,41]. 

Most of the changes in plasma metabolites after prolonged and intensive exercise reach their nadir within a few hours. Plasma deviations in many of these metabolites are still apparent, but largely abated, after one day of recovery [5,7,12,19,20,21]. The large and varied metabolite response to heavy exercise workloads reflects the physiological stress and diminished glycogen stores experienced by the participant [12,21,24,41]. 

An increasing number of studies are utilizing metabolomics to measure the influence of various nutritional interventions on metabolite perturbations during recovery from prolonged and intensive exercise [5,7,14,16,41,42]. Metabolomics is ideally suited to measuring the impact of nutritional interventions during acute exercise by simultaneously measuring and identifying shifts in hundreds of metabolites from diverse pathways. Emerging data indicate that carbohydrates from both sugar beverages and fruits such as bananas, and flavonoids from food and beverage sources such as blueberries and green tea, have a large effects on the human metabolome response to intense exercise workloads [3,5,14,16,42,43]. 

Relatively few studies have investigated exercise-induced metabolite changes following acute bouts with lower durations (< 60 min) and workload volumes [23,26]. Half of these studies performed metabolomics using GC-MS or NMR analytical platforms, limiting the number of identified metabolites and the usefulness of these data. As expected, post-exercise shifts in plasma metabolite levels are modest in comparison to high exercise volume workloads due to a moderated reliance on underlying carbohydrate and lipid substrate pathways. 

More cross-sectional studies are needed to compare plasma and urine metabolite levels between sedentary and physically active individuals, and athletes from different sports. These studies could provide important information for future randomized, exercise training trials. Using a cross-sectional design, one study showed some metabolite differences between power and endurance athletic groups [15]. The athletes were not tested at the same time or in similar resting states, however, making it difficult to draw definitive conclusions. 

Few randomized, exercise training studies have been conducted to investigate potential adaptations in the human metabolome [25,27]. These two studies employed different training protocols and study designs, and performed metabolomics using NMR, limiting the usefulness of these data. Future metabolomics-based randomized exercise training studies, especially when combined with genomics and proteomics outcomes, will improve scientific understanding of the human system’s response to varying exercise workloads [44].

## 4. Materials and Methods 

This systematic review was performed according to the Preferred Reporting Items for Systematic Reviews and Meta-Analyses (PRISMA) guidelines [45] and was preregistered in the International Prospective Register of Systematic Review (PROSPERO). To systematize the search and data extraction, a free standardized electronic tool called State of the Art through Systematic Review (StArt) [46] was used. The software StArt tracked duplicated studies during extraction, and this was confirmed with manual examination by the two main reviewers. The studies were selected, extracted and included independently by two researchers (CAS and EFS), and a third independent researcher (RMA) verified the inclusion process in order to solve any disagreement between the two main researchers. 

### 4.1. Search Strategy 

The electronic search was performed from inception to November 26th, 2018 and updated on April 10th, 2019. The articles were retrieved from the following electronic databases: PubMed (via National Library of Medicine), Science Direct, SCOPUS (Elsevier) and Web of Science. The MeSH terms were selected and combined according to analysis method (metabolomics) and mandatory activity (sports OR exercise). Moreover, the search strategy was limited to humans (population of interest), English language and clinical trial studies.

### 4.2. Eligibility Criteria for Inclusion

The abstracts were first examined and evaluated for the listed criteria. Studies were selected if metabolomics were utilized to measure exercise-induced changes in metabolites in healthy study participants using serum, plasma, saliva, or urine samples. Exercise-based studies with nutrition interventions were included, but this review only included data collected from the control groups. Reviews, case reports, guidelines, theses and dissertations, conference abstracts, and studies using animal or in vitro models were not included. 

### 4.3. Data Extraction and Study Inclusion 

The following data from the selected studies were extracted: name of the first author, year of publication, characteristics of participants and groups (population, sample size, groups, gender, age, physical activity level), research design elements (type of research, exercise mode, duration, and intensity), metabolomics procedures (analytical platform, metabolite data), and summary comments. 

### 4.4. Studies Quality Assessment

The quality of the studies was assessed by two researchers (CAS and EFS) using a scoring system created for this analysis (see Table 4 and Figure 2).

## 5. Conclusions and Future Directions 

The first decade of metabolomics-based exercise studies, especially those utilizing sensitive LC-MS/MS analytical platforms with large chemical standards libraries and rigorous bioinformatics support, provided useful systems biology information on the biochemical mechanisms underlying exercise-induced effects on metabolism [13,47]. This area of scientific endeavor is still emerging, and much remains to be discovered, especially in the areas of the metabolite response to acute and chronic moderate exercise workloads. The sensitivity of the analytical platforms will continue to improve, expanding the number of small molecule metabolites that can be detected. These improvements in technology, coupled with improved quality control, bioinformatics support, the expansion of biochemical standards, and an emphasis on larger study groups of both genders, will improve the identification and quantitation of currently known and unknown metabolites in a variety of human matrixes. More emphasis is needed on the influence of activity reduction and physical inactivity on metabolite shifts. These improvements in study design and methodology will broaden our understanding of the influence of acute and chronic exercise on the human metabolome. An increasing number of studies, including the National Institutes of Health project, ’Molecular Transducers of Physical Activity in Humans’, will combine metabolomics with genetics, epigenetics, lipidomics, and proteomics to examine all aspects of the physiological, biochemical, and molecular response to both aerobic- and resistance-based exercise training interventions [44].

## Figures and Tables

**Figure 1 metabolites-09-00164-f001:**
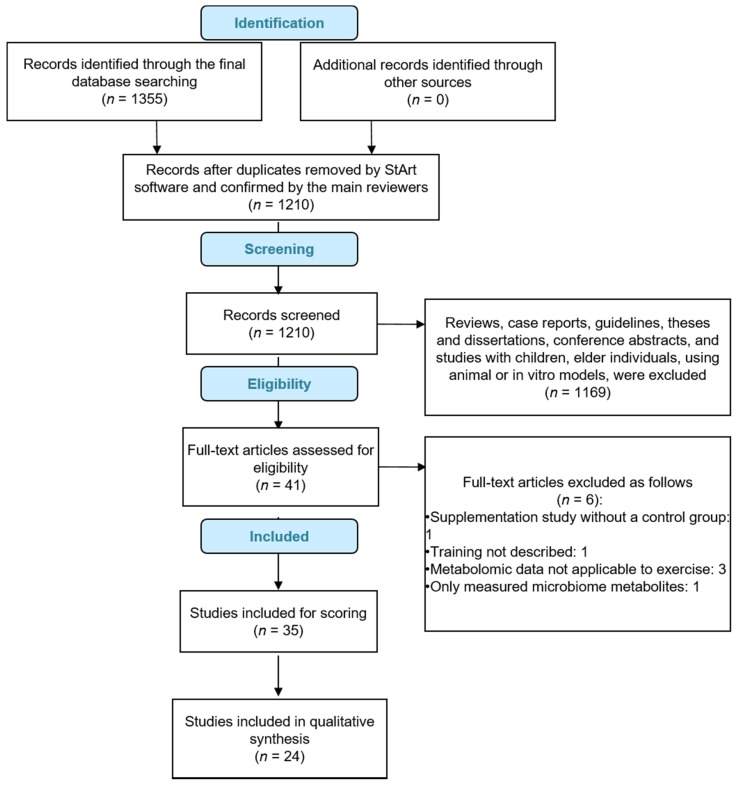
Outcomes of review flow diagram.

**Figure 2 metabolites-09-00164-f002:**
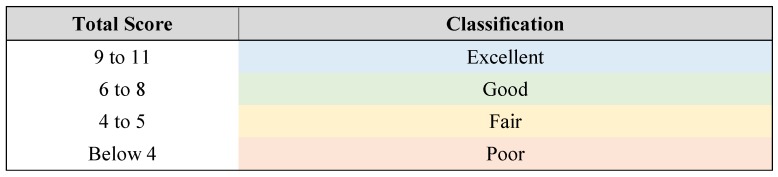
Classification of studies based on the total score.

**Table 1 metabolites-09-00164-t001:** Studies classification according to score system.

Investigators, Year Published	Research Design		Methodology		Novelty	Final Score	Classification
	Subjects Number	Studies Characteristics	Analysis Methods	Statistical Support			
Nieman et al., 2015 [5]	2	2	3	2	2	11	Excellent
Jacobs et al., 2014 [6]	2	2	3	2	2	11	Excellent
Nieman et al., 2014 [7]	2	2	3	1	2	10	Excellent
Nieman et al., 2017 [8]	2	1	3	2	2	10	Excellent
Davison et al., 2018 [9]	2	2	3	1	1	9	Excellent
Hodgson et al., 2012 [10]	0	2	3	2	2	9	Excellent
Karl et al., 2017 [11]	0	1	3	2	2	8	Good
Lehman et al., 2010 [12]	0	1	3	2	2	8	Good
Lewis et al., 2010 [13]	2	1	3	0	2	8	Good
Nieman et al., 2013 [14]	0	2	3	1	2	8	Good
Al-Khelaifi et al., 2018 [15]	2	0	3	2	1	8	Good
Knab at al., 2013 [16]	0	2	1	2	2	7	Good
Manaf et al., 2018 [17]	0	1	3	2	1	7	Good
Messier et al., 2017 [18]	2	1	1	2	1	7	Good
Nieman et al., 2013 [19]	0	1	3	1	2	7	Good
Nieman et al., 2014 [20]	0	1	3	1	2	7	Good
Ra et al., 2014 [21]	2	1	1	2	1	7	Good
Stander et al., 2018 [22]	2	1	1	2	1	7	Good
Danaher et al., 2015 [23]	0	1	1	2	2	6	Good
Howe et al., 2018 [24]	0	0	3	2	1	6	Good
Neal et al., 2013 [25]	0	1	1	2	2	6	Good
Peake et al., 2014 [26]	0	2	1	1	2	6	Good
Pechlivanis et al., 2013 [27]	0	1	1	2	2	6	Good
Zafeiridis et al., 2016 [28]	0	1	1	2	2	6	Good
Muhsen Ali et al., 2016 [29]	0	0	3	2	0	5	Fair
Castro et al., 2019 [30]	2	1	1	1	0	5	Fair
Enea et al., 2010 [31]	0	1	1	2	1	5	Fair
Andersson Hall et al., 2015 [32]	0	1	1	2	1	5	Fair
Pechlivanis et al., 2010 [33]	0	1	1	2	1	5	Fair
Wang et al., 2015 [34]	0	1	1	2	1	5	Fair
Yan et al., 2009 [35]	0	1	1	2	1	5	Fair
Prado et al., 2017 [36]	2	0	2	0	0	4	Fair
Sun et al., 2017 [37]	0	0	1	2	1	4	Fair
Berton et al., 2017 [38]	0	1	1	0	1	3	Poor
Valério et al., 2018 [39]	0	1	1	0	1	3	Poor

**Table 2 metabolites-09-00164-t002:** High-intensity and long-duration studies.

Investigators Year Published	Study Population	Analytical Platform/Matrix	Research Design	Key Findings Exercise Effect Separate from Other Interventions	Intensity
Nieman et al., 2015 [5]	20 male cyclists (aged 39.2 ± 1.9 years)	UPLC-MS/MS; Plasma	Randomized, cross-over design; three trials of a 75-km cycling protocol ingesting: water only, bananas and water, pears and water (2-week washout); blood samples timepoints: pre- and post-exercise (0-h, 1.5-h, 21-h)	509 metabolites were chemically identified; ↑ ratio > 2-fold: 107 metabolites increased in the water only trial (exercise effects); ↑ ratio > 5-fold: 35 metabolites increased, all from the lipid super pathway, all significantly elevated 1.5-h post exercise, 8 only remained after 21-h post-exercise.	High-intensity, long-duration
Nieman et al., 2014* [7]	19 male cyclists (aged 38.06 ± 1.6 years)	GC-MS and UHPLC-MS/MS; Plasma	Randomized, cross-over design; two trials of a 75-km cycling protocol with pistachio or no pistachio supplementation (2-week washout); blood samples timepoints: pre- and post-exercise (0-h, 1.5-h, 21-h)	423 metabolites were chemically identified; Exercise increased 167 metabolites; All but 26 of these metabolites were related to ↑ lipid and carnitine metabolism, with the largest fold changes seen for ketones, dicarboxylate fatty acids, and long chain fatty acids.	High-intensity, long-duration
Nieman et al., 2017 [8]	24 trained male runners (aged 36.5 ± 1.8 years)	GC-MS and UHPLC-MS/MS; Plasma	Repeated measures, ANOVA analysis, one group design; blood samples collected pre- and post-exercise (0-h), one bout run to exhaustion at 70%VO_2max_	209 chemically identified metabolites changed with exercise, especially long and medium-chain ↑ fatty acids, ↑ fatty acids oxidation products (dicarboxylate and monohydroxy fatty acids and acylcarnitines), and ↑ ketone bodies. Minor relationship with ↑ IL-6.	High-intensity, long-duration
Davison et al., 2018 [9]	24 healthy males (aged 28 ± 5 years)	LC-MS; Serum	Double-blind, randomized, cross-over design; 60-min run 75% VO_2max_ in hypoxia (FiO_2_ = 0.16%) (hypoxia chamber) and normoxia (FiO_2_ = 0.21%) (1-week washout); blood samples timepoints: pre- (after 30-min rest in hypoxia, normoxia), post-exercise (0 h, 3-h)	27 metabolites, identified using internet databases, changed with exercise; Most related to ↑ lipid metabolism (several acylcarnitines molecules identified) and purine metabolism [↑adenine, ↑adenosine and ↓ (3 h after recovery) hypoxanthine]; ↑ 4.3-fold increase in 18 acylcarnitines post-exercise, above pre-exercise at 3-h recovery.	High-intensity, long-duration
Lehman et al., 2010 [12]	Healthy subjects; 1st study: *n* = 13 (32.6 ± 6.1 years) 2nd study: *n* = 8 (30.9 ± 5.8 years)	UPLC-qTOF-MS; Plasma	Parallel group design; 1st study: treadmill run 60min at 75% VO_2_, blood samples timepoints: pre- and post-exercise (0-h, 3-h, 24-h); 2nd study: treadmill run > 120 min at 70%V_IAT_, blood samples timepoints: pre- (1h 45 min after breakfast) and post-exercise (0-h, 3-h, 24-h)	10 metabolites, chemically identified, characterized the separation between the timepoints; Most part non-esterified free fatty acids; ↑ 9-fold increases in acylcarnitines.	High-intensity, long-duration
Lewis et al., 2010 [13]	25 amateur runners (aged 42 ± 9 years)	LC-MS; Plasma	Repeated measures, one group; Boston Marathon; blood samples time points: pre- and post-marathon	Metabolites chemically identified; ↑ in glycolysis, lipolysis, adenine nucleotide catabolism, and amino acid catabolism; ↑ indicators of glycogenolysis (glucose-6-phosphate and 3-phosphoglycerate), and small molecules that reflect oxidative stress (allantoin), and that modulate insulin sensitivity (niacinamide)	High-intensity, long-duration
Nieman et al., 2013 [14]	35 long-distance male runners (supplemented group: aged 33.7 ± 6.8 years; placebo: aged 35.2 ± 8.7 years)	GC-MS and UHPLC-MS/MS; Serum	Double-blind, parallel group design; 2-week supplementation (polyphenol-enriched protein) followed by a 3-day intensified exercise (2.5-h at 70%VO_2max_ bouts); blood samples timepoints: pre- and post- 14-day supplementation, and immediately and 14-h after the 3rd day of running	324 chemically identified metabolites that changed with 3-day period of exercise; ↑ metabolites related to fatty acid oxidation and ketogenesis including free fatty acids, acylcarnitines, 3-hydroxy-fatty acids, and dicarboxylic acids, amino acid and carbohydrate metabolism, energy production, nucleotides, and cofactors and vitamins.	High-intensity, long-duration
Knab et al., 2013 [16]	9 elite male sprint and middle-distance swim athletes; 7 control subjects (healthy and exercised less than 150 min/week) (aged 24.6 ± 0.7 years)	GC-MS; Serum	Randomized, crossover design, 10-day supplementation with juice (8 fl oz pre- and post-training) or non-juice, 10-d practice of 2-h swimming, approximately 5500-m swim interval training (3-week washout). Blood samples timepoints: pre- and post- each 10-days supplementation period and post-exercise (0-h)	325 metabolites were chemically identified; No effects of juice on exercise-induced measures; ↑ Oxidative stress and ↓ antioxidant capacity in swimmers group compared to nonathletic control group; Metabolites that differed mostly related to substrate utilization and supplements used by the swimmers. Pre and post-exercise small but significant shift in metabolites related to substrate utilization: pyruvic acid, propanoic acid, d-fructose, mannose, n-acetylglutamine, norleucine, alloisoleucine, and d-glucuronic acid.	High-intensity, long-duration
Manaf et al., 2018 [17]	18 healthy and recreationally active males (aged 24.7 ± 4.8 years)	LC-MS; Plasma	Repeated measures, ANOVA analysis, one group design; time-to-exhaustion (81-min) cycling test at a workload 3 mM/l lactate; blood samples timepoints: pre-exercise, during exercise (10-min, before fatigue), point of exhaustion (immediately after fatigue), post-exercise (20-min after fatigue)	80 metabolites identified using internet databases; 68 metabolites changed during exercise; ↑ Free-fatty acids and ↓ tryptophan contributed to differences in plasma metabolome at fatigue.	High-intensity, long-duration
Messier et al., 2017 [18]	20 healthy male (aged 39 ± 4.3 years)	1H NMR; Plasma	Cross-over design; cycling 60-min at ventilatory threshold 1 at 70 rpm, at sea level and above 2150 m of the sea level (2-week washout); blood samples timepoints: pre- and post-exercise (0-h)	18 metabolites identified using internet databases; ↓ glucose and free amino acid levels; No differences in lipid metabolism between altitudes; Fuel shift from lipid oxidation to carbohydrate oxidation at 2150 above sea level.	High-intensity, long-duration
Nieman et al., 2013 [19]	15 runners (7 males, 8 females) (aged 35.2 ± 8.7 years)	GC-MS and UHPLC-MS/MS; Serum	Cross-sectional design, 3-day period exercise (2.5 h per day running bouts at approximately 70% VO_2max_); blood samples timepoints: pre- and post-exercise (0-h, 14-h)	Metabolites chemically identified; ↑ ≥ 2-fold increases in 75 metabolites immediately post 3-day exercise period, 22 related to lipid/carnitine metabolism, 13 related to amino acid/peptide metabolism, 4 to hemoglobin/porphyrin metabolism, and 3 to Krebs cycle intermediates. After 14-h recovery: 50 of 75 metabolites still elevated. ↓ 22 metabolites post-exercise related to lysolipid and bile acid metabolism.	High-intensity, long-duration
Nieman et al., 2014* [20]	19 male cyclists (aged 38.06 ± 1.6 years)	GC-MS and UHPLC-MS/MS; Plasma	Repeated measures, ANOVA analysis, one group design; blood samples timepoints: pre- and post-exercise (0-h, 1.5-h, 21-h); 75-km cycling protocol	221 chemically identified metabolites changed with exercise; all but 26 related to ↑ lipid and carnitine metabolism; largest fold changes seen for ↑ ketones, dicarboxylate fatty acids, and long chain fatty acids.	High-intensity, long-duration
Ra et al., 2014 [21]	37 male soccer players (aged 20.6 ± 0.04 years)	CE-TOFMS; Saliva	Repeated measures, ANOVA analysis, one group design; 3-day game program (90-min per day); saliva samples timepoints: pre-exercise (1-month before) and post-exercise (24-h after)	144 metabolites chemically identified; ↑12 metabolites (e.g., 3-methylhistidine, glucose 1- and 6-phosphate, taurine, amino acids) related to muscle catabolism, glucose metabolism, lipid metabolism, amino acid metabolism and energy metabolism.	High-intensity, long-duration
Stander et al., 2018 [22]	31 recreational marathon athletes (19 males and 12 females) (aged 41 ± 12 years)	GC-TOFMS; Serum	Repeated measures, ANOVA analysis, one group design; 42-km marathon; blood samples timepoints: pre- and post-marathon (0-h)	70 metabolites chemically identified; ↑ carbohydrates, fatty acids, tricarboxylic acid cycle intermediates, ketones, and ↓ amino acids; ↑odd-chain fatty acids and α-hydroxy acids.	High-intensity, long-duration
Howe et al., 2018 [24]	9 male ultramarathon runners (aged 34 ± 7 years)	HILIC-MS; Plasma	Repeated measures, ANOVA analysis, one group design; 80.5-km treadmill simulated ultramarathon run; blood samples timepoints: pre- and post-exercise (0-h)	446 metabolites chemically identified; ↓ amino acids metabolism post-80.5 km; ↑ in the formation of medium-chain unsaturated, partially oxidized fatty acids and conjugates of fatty acids with carnitines.	High-intensity, long-duration

UPLC-MS: ultra-performance liquid chromatography mass spectrometry; UHPLC-MS: ultra-high-performance liquid chromatography mass spectrometry; GC-MS: gas chromatography mass spectrometry; LC-MS: liquid chromatography mass spectrometry; UHPLC/Q-TOF MS: ultra-high-performance liquid chromatography quadrupole time-of-flight mass spectrometry; 1H NMR: proton nuclear magnetic resonance; CE-TOFMS: capillary electrophoresis time-of-flight mass spectrometry; HILIC-MS: hydrophilic interaction chromatography mass spectrometry; VO_2max_ = maximal oxygen uptake; FiO_2_ = fraction of inspired oxygen; V_IAT_ = velocity at individual anaerobic threshold. * References [7,20] were from the same study but the data sets provided additive information.

**Table 3 metabolites-09-00164-t003:** Summaries of study characteristics and findings from nine [6,10,11,15,23,25,26,27,28] studies using other types of exercise designs.

Investigators Year Published	Study Population	Analytical Platform/Matrix	Research Design	Key Findings Exercise Effect Separate from Other Interventions	Intensity
Danaher et al., 2015 [23]	7 active males (aged 22.9 ± 5.0 years)	GC-MS; Plasma	Randomized, cross-over design; two supramaximal low volume high-intensity exercise protocols (1-week washout) (HIE); (1) HIE_150%_: 30 × 20 s cycling at 150% VO_2peak_, 40 s rest (348 ± 27W); (2) HIE_300%_: 30x 10s cycling at 300% VO_2peak_, 50 s rest (697 ± 54 W); blood samples timepoints: pre- and post-exercise (0-h, 1-h)	55 chemically identified metabolites detected; HIE300% produced greater metabolic perturbations compared to HIE150%; Changes more pronounced during recovery than exercise, with ↑ glycolytic pathway and fatty acids and lipid metabolism.	High-intensity, short-duration
Zafeiridis et al., 2016 [28]	9 healthy young men (aged 20.5 ± 0.7 years). Soccer training 4−5 times per week.	1H NMR; Plasma	Randomized, cross-over design; three running protocols (2-week washout): intense continuous (18-min, 80% of maximum aerobic velocity (MAV)), long-interval (29-min, 3 min at 95% of MAV, 3 min recovery at 35% of MAV) and short-interval (18-min, 30 s at 110% of MAV, 30 s recovery at 50% of MAV); blood sample timepoints: pre- and post-exercise (5-min).	17 metabolites identified using internet databases; No detectable difference in metabolites; ↑ carbohydrate/lipid metabolism and activation of the TCA cycle in all three protocols.	High-intensity, short-duration
Jacobs et al., 2014 [6]	19 healthy physically active males (aged 21 ± 2 years)	GC-MS and LC-MS/MS; Plasma	Double-blind, randomized, cross-over design; 6-day supplementation with decaffeinated green tea or placebo ingestion (28-day washout) 2-h before a 30 min cycle exercise at 55%_VO2max_	152 chemically identified metabolites changed with exercise; ↑ metabolites related to adrenergic and energy metabolism (e.g., lactate, pyruvate, malate, succinate, glycerol, cortisol); ↓ 2-hydrxobutyrate.	Moderate-intensity, short-duration
Hodgson et al., 2012 [10]	27 healthy physically active males (aged 22 ± 5 years)	GC-MS and LC-MS/MS; Plasma	Double-blind, randomized, parallel design; 7-day supplementation with caffeinated green tea or placebo ingestion 2-h before 60-min cycle exercise at 50%VO_2max_	238 metabolites chemically detected changed with exercise; ↑ ratio > 2: lactate, pyruvate, succinate, noradrenaline and glycerol; ↓ 2-hydroxybutyrate, trans-4-hydroxyproline, mannose, certain triacylglycerides (TAGs) and nicotinamide.	Moderate-intensity, long-duration
Karl et al., 2017 [11]	25 male highly trained soldiers (aged 19.0 ± 1.0 years)	UPLC-MS/MS; Plasma	Double-blind, randomized, parallel design; 4-day, 51-km cross-country ski march carrying 45 kg pack; blood sample timepoints: pre- and post-exercise (early completers: 8 to 10-h or late completers: 2 to 3-h).	478 chemically identified metabolites changed pre- and post-exercise ↑ 88% of the free fatty acids; ↑ 91% of the acylcarnitines; ↓ 88% of the mono- and diacylglycerols detected within lipid metabolism pathways; Smaller ↑ 75% of the tricarboxylic acid cycle intermediates; ↑ 50% of the branched chain amino acid metabolites	Moderate-intensity, long-duration
Peake et al., 2014 [26]	10 well-trained male cyclists and triathletes (aged 33.2 ± 6.7 years)	GC-MS; Plasma	Randomized, cross-over design; HIIT (60-min, ≈ 82% VO_2max,_) and a moderate-intensity continuous exercise (MOD) (61-min, ≈ 67% VO_2max_); blood samples timepoints: pre- and post-exercise (0-h, 1-h, 2-h).	49 metabolites chemically identified; 29 changed with exercise (11 changed with both HIIT and MOD; 13 changed with HIIT only; 5 changed with MOD only); ↑ in carbohydrate oxidation and ↓ in fat oxidation in HIIT exercise compared to MOD; Glucose and lactate higher at 0-h in HIIT compared to MOD.	High and moderate-intensity, long-duration
Al-Khelaifi et al., 2018 [15]	191 elite athletes (171 males, 20 females)	UPLC-MS/MS; Serum	Cross-sectional design using elite athletes from various sport disciplines being monitored for doping; blood samples collected IN or OUT competition (1 timepoint)	Metabolites chemically identified; ↑ Oxidative stress common to both high-power and high-endurance sports alike; ↑ steroids and polyamine pathways more prominent in endurance; ↑ sterols, adenine-containing purines, and energy metabolites most evident with power.	Cross-section elite athletes
Neal et al., 2013 [25]	12 male cyclists (aged 36 ± 6 years)	1H NMR; Urine	Randomized, cross-over design; 6-week training of polarized training-intensity (80% low intensity, 0% moderate-intensity, 20% high-intensity) and a training-intensity distribution (57% low intensity, 43% moderate-intensity, 0% high-intensity) (4-week washout); urine samples timepoints: pre- and post- each training period.	Method used to identify metabolites not reported; metabolites identified as ↓ hippuric acid, ↑ creatinine, ↑ dimethylamine, ↑ 3-methylxanthine, ↓ hypoxanthine.	Chronic training, Low, moderate and high-intensity
Pechlivanis et al., 2013 [27]	14 young moderately trained healthy males (aged 21 ± 2 years)	1H NMR; Serum	Randomized, parallel group design; two 8-week programs (3 sessions/week), two and three sets of two 80-m maximal runs (interval between runs: group A = 10 s; group B = 1 min), 20 min interval between sets; blood timepoints: pre- and post-training.	18 chemically identified metabolites changed after training period; separation after training mainly due to ↓ lactate, ↓ pyruvate, ↑ methylguanidine, ↑ citrate, ↑ glucose, ↑ valine, ↑ taurine, ↑ trimethylamine N-oxide, ↑ choline-containing compounds, ↑ histidines, ↑ acetoacetate/acetone, ↓ glycoprotein acetyls, and ↓ lipids; no significant difference between training intervals.	Chronic training, high-intensity

GC-MS: gas chromatography mass spectrometry; LC-MS: liquid chromatography mass spectrometry; UPLC-MS: ultra-performance liquid chromatography mass spectrometry; 1H NMR: proton nuclear magnetic ressonance; UPLC-MS: ultra-performance liquid chromatography mass spectrometry; HIE = high-intensity exercise; HIE150% = high-intensity exercise at 150% of VO2 peak; HIE300% = high-intensity exercise at 300% of VO2 peak; W = watts; MAV = maximum aerobic velocity; TCA = tricarboxylic acid cycle; HIIT: high-intensity interval training; MOD = moderate-intensity continous exercise.

**Table 4 metabolites-09-00164-t004:** Score setting for metabolomic studies quality assessment.

Score Setting			
Section	Maximum Score	Aspects	Score Attribution
**Research Design**	2	Number of Participants	Parallel Studies 0 – N < 20 2 – N > 20 Crossover Studies 0 – N < 13 2 – N > 13
	2	Study Characteristics	Randomized control group Proper matrix > 2 timepoints data collection Duration ≥ 3 week (chronic studies only) 0—None of the previous items 1—At least 2 of the first 3 criteria listed 3—All 3 of the first 3 criteria listed
**Methodology**	3	Analysis Methods	3—LC-MS/MS with extensive standards 1—NMR 1H, limited standards 1—GC-MS, limited standards
	2	Statistical Support	0—simple univariate statistics 1—Univariate statistics + additional analyses to sort and group the data, and to control for confounding factors 2—Univariate statistics + PCA, OPLS-DA, PLS-DA, or similar advanced bioinformatics procedures
**Novelty**	2		New information in the literature

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
