# Peer review of "Metabolomics-Based Studies Assessing Exercise-Induced Alterations of the Human Metabolome: A Systematic Review"

_metabolites, 2019, doi:10.3390/metabo9080164_

Round 1

Reviewer 1 Report

The authors conducted a systematic literature review and summarised the human metabolome response to exercise with varied intensities and durations using a scoring system for quality assessment of the previously published literature in the field. The authors gathered and provided detailed information on the study design, for example, characteristics of the cohorts and exercise/training methods, types of biological samples (blood, saliva, urine), metabolomics analytical platforms (LC-MS/MS, NMR and GC/MS), statistics robustness and key metabolites findings by virtue of exercise. Overall, the review is descriptive, provides insights into the current state of knowledge of exercise effects on the human metabolome and may serve as a foundation for the scientific community for designing and conducting future informed research. A few minor suggestions are included below for the authors' consideration:

Line 59-61: it would be helpful to provide further information specifying the exclusion criteria for the selection of studies that are included in Table 1, in addition to stating "...excluded for not meeting analysis criteria after review of the abstracts" only. Please also note on line 60 and 61, the numbers are inconsistent among 41, 5 and 35.

Page 4, Figure 1: two queries here: 1) duplicated records are removed; please specify how the duplicates were defined and subsequently removed; 2) please also provide the exclusion criteria for the records (n=1169) in the text box in the Figure. Such information will serve the readers well without referring to other sections of the review for information and for consistency.

Table 2: several studies in Table 2 seem to be conducted by the same group of authors using a similar design i.e. Nieman et al. Did samples analysed in these studies overlap? what are considerations and limitations when the samples are repeatedly used for varying research questions? Please address these.

Table 4: Detailed score classification (i.e. for level 0, 1, 2, 3) needs to be clarified for each component/aspect and included in the "score attribution" column of the Table.

Tables: key abbreviations used in a Table should be clarified underneath the Table.

Typos in the text need to be corrected, e.g. CG-MS to GC-MS in Table 4.

Reviewer 2 Report

The overall subject of the study is likely to be of broad interest and the authors have done a good job collecting the relevant literature.  I am not familiar with the StArt system described in the manuscript and used for selecting the initial set of article, but a random review of references supplied appears to cite relevant literature. A review and light editing of prose and language should help.

My only concern is related to the "scoring system" described in the methodology.  Table 4 describes the scoring system, which appears somewhat arbitrary to me.  As one example, the rows related to "Methodology" add to a total of 5 points from 11 points possible. This weighting makes it possible for a study to go from excellent to good, or even fair very easily.

The weighting of the Methodology is reasonable, but how it is evaluated seems to beg the question - why is an NMR study receives a "1" and an LC-MS study receives a "3"?  Some NMR studies report on meticulous quantitative evaluation of metabolites, while some LC-MS studies are have been shown to contain many spurious identifications.  Additionally, it is unclear as to why PCA and PLS-DA have been distinguished as deserving of additional score in the statistical support row -  there are many other approaches that could be used.

I would like to see the authors explain to the audience of the article their selection of weights for the analysis method.

With regard to the statistical support, perhaps the choice is made on the basis of the totality of all methods used in the articles surveyed. In other words, the authors only encountered the methods mentioned in the table.  If that is the case, I would like to see that explicitly stated.  If this is not the case, then an explanation will be helpful.

Reviewer 3 Report

Th authors provide a systematic review of metabolomic studies related to human exercise interventions. Overall, it is a concise report that highlights "general" trends in the field in terms of study designs, type of exercise regimes and major platforms used for metabolome analysis - however, it is not a critical review per se in my opinion. In order to rank quality of contributions, the authors developed a standardized scoring scheme as selection criteria for papers considered as high quality for consideration in their review . However, this process results in exclusion of the vast majority of relevant papers in the literature, including many novel yet smaller scale investigations reported to date. For example, the authors seem to focus exclusively on blood or urine investigations despite recent innovative developments in analyzing skeletal muscle metabolome changes following exercise (Anal. Chem. 2019, 91: 4709; J. Proteome Res. 2016, 15: 499). Additionally, step reduction could be considered as an intriguing study to explore the role of physical inactivity on human health that is recommended to be included in their review perhaps as future directions (Metabolites 2019, 9: 134), as well as adaptive responses to exercise training (Electrophoresis 2015, 36: 2226). Also, the authors are recommended to differentiate between targeted analysis of panels of metabolites (most studies published to date), and truly non-targeted or discovery-based metabolomic studies, as well as the challenges for unknown metabolite identification. Other major challenges in the field include lack of quantitative measurements (absolute concentrations or reference ranges) for better laboratory comparisons, frequent use of of single sex/underpowered studies, as well as the importance of stringent quality control and appropriate statistical analyses to avoid false discoveries, including replication/validation. It is recommended that the authors consider some of these suggestions in order to improve the quality of their review that provides readers more insights into new directions and ongoing challenges in the field.

Reviewer 4 Report

Metabolites MDPI, Metabolites-558974

Summary: This review discusses the current knowledge surrounding metabolic adaptations to different lengths and intensities of exercise as it relates to the high throughput method of metabolomics. It was noted that the majority or research has been conducted in the area of acute effects of long duration (>60mins) or high intensity (>60% of oxygen uptake and heart rate reserve), which showed significant increases in beta oxidation pathway metabolites within a few hours of completion of exercise. These values returned to baseline levels within 24 hours. The review goes on to touch on the remaining studies and how short duration or low-moderate intensity exercise impacted the metabolome. The review closes by advocating for the need of more studies focusing on sedentary and physically active humans as well as study participant randomization. It also advocates for the use of multi-omic studies that encompass multiple layers of molecular mechanisms and signaling in order to understand the net effect of exercise on the human body.

Major Point: This is an interesting and potentially valuable review of the hot field of metabolomics as it relates to exercise-mediated adaptations. The tables and figures breaking down literature selection and findings was helpful and appropriate. Particularly, the level of detail in Table 2 and 3 outlining specifics of study populations (male vs. female, athlete type, age, analytical platform, research design, etc) was appreciated. Such a review has potential to be impactful pending edits that more clearly define metabolite shift direction and how these shifts could/should be interpreted. No major revisions suggested.

Minor Points:

1.      Consider making a separate column in Table 2 and 3 for tissue type. Or (since the tables fit nicely on the page as is) consider creating a subheading that separates by tissue. Tissue type is such a critical part of the interpretation of metabolites. 

2.      In the text (particularly points A-G), it would be helpful to have more discussion of the direction of changes in metabolites (increased, decreased) and in which tissues rather than just reporting that there were changes. This would substantially strengthen the review.

3.      The first paragraph of the discussion uses a fair amount of recycled text from the abstract and other places in the review. Original text would be appreciated by the audience.

4.      The discussion would be strengthened by including more information about the direction of metabolite shift and what this may mean. For example, the review mentions that supplementation appears to “have a large effect on the human metabolome response to intense exercise workloads [3,5,14,16,42,43]”, yet this effect is not mentioned. More discussion about what was found, direction of metabolite shift, and interpretation would be valuable throughout the discussion.
